# State-Level Health Disparity Is Associated with Sarcoidosis Mortality

**DOI:** 10.3390/jcm10112366

**Published:** 2021-05-27

**Authors:** Yu-Che Lee, Ko-Yun Chang, Mehdi Mirsaeidi

**Affiliations:** 1Department of Medicine, University at Buffalo-Catholic Health System, Buffalo, NY 14220, USA; yuchelee@buffalo.edu; 2Division of Chest Medicine, Taichung Veterans General Hospital, Taichung 40705, Taiwan; karen71531@gmail.com; 3Division of Pulmonary and Critical Care Medicine, University of Miami Miller School of Medicine, Miami, FL 33146, USA; 4Section of Pulmonary Medicine, Miami Veterans Affairs Healthcare System, Miami, FL 33125, USA

**Keywords:** USA, health disparity, sarcoidosis, mortality

## Abstract

Background: Sarcoidosis is associated with significant morbidity and rising health care utilization, which contribute to the health care burden and disease outcome. In the United States (US), evaluation of sarcoidosis mortality by individual states has not been investigated. Methods: We examined sarcoidosis mortality data for 1999–2018 from the Centers for Disease Control and Prevention (CDC). America’s Health Rankings (AHR) assesses the nation’s health on a state-by-state basis to determine state health rankings. The numbers of certified Sarcoidosis Clinics within the US were obtained from World Association for Sarcoidosis and Other Granulomatous Disorders (WASOG) and Foundation for Sarcoidosis Research (FSR). The associations between sarcoidosis mortality and state health disparities were calculated by linear regression analyses. Results: From 1999 to 2018, the mean age-adjusted mortality rate (AAMR) in all populations, African Americans and European Americans were 2.9, 14.8, and 1.4 per 1,000,000 population, respectively. South Carolina had the highest AAMR for all populations (6.6/1,000,000) and African Americans (20.8/1,000,000). Both Utah and Vermont had the highest AAMR for European Americans (2.6/1,000,000). New York State and South Atlantic had the largest numbers of FSR-WASOG Sarcoidosis Clinics (6 and 13, respectively). States with better health rankings were significantly associated with lower AAMR in all population (R^2^ = 0.170, *p* = 0.003) but with higher AAMR in European Americans (R^2^ = 0.223, *p* < 0.001). Conclusions: There are significant variations in sarcoidosis mortality within the US. Sarcoidosis mortality was strongly associated with state health disparities. The current study suggests sarcoidosis mortality could be an indicator to reflect the state-level health care disparities in the US.

## 1. Introduction

Sarcoidosis is a multisystem granulomatous disease of unknown origin that mainly affects lungs, lymph nodes, skin, and eyes [1]. Racial and geographic disparities in sarcoidosis epidemiology, disease course, and prognosis have been well described, indicating African Americans and northern European countries are most affected worldwide [2]. In the United States (US), the differences in sarcoidosis incidence and mortality rates between races have been reported with African Americans consistently having higher rates than European Americans. African Americans have a 3–5 fold greater incidence of sarcoidosis and a 10–12 fold greater mortality rate compare to European Americans [3,4]. Geographic variabilities of sarcoidosis-related mortality in the US have also been documented with the highest rate in the southern region for all population, the Midwest for African Americans females, and New England states for European Americans [3,4,5,6].

Sarcoidosis is associated with significant morbidity and rising health care utilization, which contribute to the health care burden and disease outcome [7,8]. Despite previous studies discussing the epidemiology and outcome of sarcoidosis, no study has evaluated the role of the state-level health disparity in mortality. We hypothesized that sarcoidosis-related mortality will be correlated with state health rankings and the accessibility of sarcoidosis medical centers, which reflect the health care disparities between states. We evaluated the association of sarcoidosis-related mortality, state health rankings, and numbers of sarcoidosis medical centers to provide a general overview of the relationship between mortality and health care disparities in the US.

## 2. Methods

### 2.1. Data Resources and Study Design

Mortality data for 1999–2018 were obtained from the Centers for Disease Control and Prevention, Wide-ranging Online Data for Epidemiologic Research (CDC WONDER). Data are based on death certificates for US residents and sarcoidosis-related mortality was defined as all decedents with International Classification of Diseases, 10th Revision (ICD-10) D86 codes as the underlying cause of death. Age-adjusted mortality rates (AAMR) per 1,000,000 population per year were generated from CDC WONDER based on weighting averages of the age-specific death rates and comparing with relative mortality risk among 2000 US standard populations.

America’s Health Rankings (AHR) was built in 1990 and partnered with the United Health Foundation and the American Public Health Association. AHR assesses the nation’s health on a state-by-state basis with different measures to determine the 50 US states’ health rankings annually, with “1” being the healthiest state and “50” the least healthy state. The state health rankings are a composite index of health measures and calculated by adding each ranked component multiplied by its assigned weight in five core measures (four groups of health Determinants and one health Outcomes): 25% for Behaviors (drug death, excessive drinking, high school graduation, obesity, physical inactivity, smoking), 22.5% for Community & Environment (air pollution, children in poverty, infectious disease, chlamydia, pertussis, salmonella, occupational fatalities, violent crime), 12.5% for Policy (immunization, public health funding, uninsured), 15% for Clinical Care (dentists, low birth weight, mental health providers, preventable hospitalizations, primary care physicians), and 25% for Outcomes (cancer deaths, cardiovascular deaths, diabetes, disparity in mental status, frequent in mental & physical distress, infant mortality, premature death). We obtained twenty-year averages of Overall health value, All Determinants value, and All Outcomes value from AHR and calculated the state health rankings for 1999–2018 respectively.

World Association for Sarcoidosis and Other Granulomatous Disorders (WASOG) is a unique organization for the diagnosis and treatment of sarcoidosis and related conditions. WASOG partnered with the Foundation for Sarcoidosis Research (FSR) to provide a platform recognizing institutions as certified Sarcoidosis Clinics. We obtained the numbers of certified Sarcoidosis Clinics within the US from WASOG and divided them into 2 groups (states with and without Sarcoidosis Clinics) and 9 geographic divisions defined by the United States Census Bureau.

### 2.2. Statistical Analysis

We thereafter generated scatterplots of Overall health rankings, All Determinants health rankings, All Outcome health rankings by state, and Sarcoidosis Clinics numbers versus Sarcoidosis AAMR, respectively. Univariate linear regression analyses were performed using the following formulas to evaluate the association between sarcoidosis AAMR, state health rankings, and Sarcoidosis Clinics: Sarcoidosis AAMR = Overall Health Rankings × β + α; Sarcoidosis AAMR = All Determinants health rankings × β + α; Sarcoidosis AAMR = All Outcome health rankings × β + α; Sarcoidosis AAMR = States with or without Sarcoidosis Clinics × β + α; Sarcoidosis AAMR = Sarcoidosis Clinics numbers × β + α. Statistical analyses and data management were performed using Statistics Software *SAS*^®^ (SAS Institute, Cary, NC, USA) and Microsoft Excel (Redmond, WA, USA), with *p*-Values < 0.05 in two-sided *T*-tests to indicate statistical significance.

## 3. Results

### 3.1. Sarcoidosis Mortality Rates by State

From 1999 to 2018, a total of 50,567,774 deaths occurred in the US, and sarcoidosis was documented as the underlying cause of death in 18,877 decedents (10,922 for African Americans, 7765 for European Americans, 131 for Asian or Pacific Islander, and 59 for American Indian or Alaska Native). The overall 20-year sarcoidosis-related AAMR in all populations, African Americans and European Americans were 2.9, 14.8, and 1.4 per 1,000,000 population, respectively. There was a remarkable geographic variation in AAMR by race. The highest AAMR for all populations was recorded in South Carolina (6.6 per 1,000,000) and the lowest in Arizona (1.0 per 1,000,000). South Carolina also had the highest AAMR for African Americans (20.8 per 1,000,000) and Nevada had the lowest AAMR for African Americans (7.1 per 1,000,000). For European Americans, the lowest AAMR was also reported in Arizona (0.8 per 1,000,000) and the highest in both Utah and Vermont (2.6 per 1,000,000). Table 1 summarizes the results and Figure 1 shows the sarcoidosis-related AAMR in African Americans and European Americans by state for 1999–2018.

### 3.2. Association between State Heath Rankings and Sarcoidosis-Related AAMR

Table 1 and Appendix A show the health rankings for 50 states from 1999 to 2018 in three different categories: Overall health rankings, All Determinants health rankings, and All Outcomes health rankings. Among 50 states, Hawaii had both the highest Overall and All Outcomes health ranking and Mississippi had both the lowest Overall and All Outcomes health ranking. Regarding All Determinants health rankings, Vermont was the healthiest and Louisiana was the least healthy. States with better Overall, All Outcomes and All Determinants health rankings were significantly associated with lower sarcoidosis-related AAMR in all population (R^2^ = 0.170, *p* = 0.003; R^2^ = 0.286, *p* < 0.001; R^2^ = 0.095, *p* = 0.033, Figure 2) whereas an opposite association was found in European Americans (R^2^ = 0.223, *p* < 0.001; R^2^ = 0.095, *p* = 0.035; R^2^ = 0.248, *p* < 0.001, Figure 3). There was no association between health rankings and sarcoidosis-related AAMR in African Americans (R^2^ = 0.023, *p* = 0.389; R^2^ = 1.66 × 10^-5^, *p* = 0.981; R^2^ = 0.044, *p* = 0.226, Figure 4).

### 3.3. Association between Sarcoidosis Clinics and Sarcoidosis-Related AAMR

Table 1 and Table 2, and Appendix A show the numbers of Sarcoidosis Clinics and sarcoidosis-related AAMR for each state and 9 geographic divisions. Among 50 states, New York had the largest number of Sarcoidosis Clinics (6). States with Sarcoidosis Clinics had 0.9 per 1,000,000 higher AAMR compared to those without Sarcoidosis Clinics in all population (*p* = 0.018), but there was no statistical difference in European Americans (*p* = 0.08) and African Americans (*p* = 0.88). Regarding geographic divisions, South Atlantic had the largest numbers of Sarcoidosis Clinics (13), the highest AAMR (4.2 per 1,000,000) for all population, and the second highest AAMR (16.3 per 1,000,000) for African Americans. In contrast, New England had the smallest number of Sarcoidosis Clinics (3) but the highest AAMR (1.8 per 1,000,000) for European Americans. Divisions with larger numbers of Sarcoidosis Clinics were associated with higher AAMR in all population and African Americans (R^2^ = 0.454, *p* = 0.047; R^2^ = 0.447, *p* = 0.049, respectively, Appendix A). There was no association between Sarcoidosis Clinics and AAMR in European Americans (*p* = 0.99).

## 4. Discussion

This study demonstrates geographic differences in sarcoidosis-related mortality in the US for 1999–2018, and to our knowledge, the first-ever assessment to evaluate the association between sarcoidosis-related AAMR and state-level health disparities. The mapping of sarcoidosis-related AAMR shows the different distribution for all populations, African Americans and European Americans. The highest AAMR for all populations was found in South Carolina, Maryland, and Alabama whereas the lowest in Arizona, New Mexico, and Nevada. For African Americans, the highest AAMR was seen in South Carolina, North Carolina, and West Virginia but the lowest in Nevada, Arizona, and Florida. In regard to European American, Northeastern states, including Vermont, Maine, and Rhode Island, had a higher AAMR than most southern states. These geographic findings are similar to the previous studies by Swigris et al. [3] and Mirsaeidi et al. [4], who both reported that sarcoidosis-related mortality rates were greater in some Northeastern states for European Americans.

Sarcoidosis-related mortality is affected by the management of long-term complications and the effectiveness of treatments including corticosteroids, cytotoxic drugs, and biologic agents [9,10,11,12,13,14,15,16,17,18,19,20,21]. State health rankings represent the availability and affordability of medical services and therapeutic options resulting in the disparity of health care by state. We found a positive association between state health rankings and sarcoidosis-related AAMR in all populations, indicating higher mortality rates in states with worse health rankings. Our model showed the highest correlation in All Outcomes health rankings, explaining 28.6% of the variability for AAMR, followed by Overall health rankings (16.9%) and All Determinants health rankings (9.5%). This significant association can be explained by attributing better All Outcomes health rankings to greater general health care capacities, which lead to better access to state-of-art medications, a higher quality of management for sarcoidosis-related complications, and lower mortality rates. Racial population distribution in the US, environmental exposures, occupational exposures, socioeconomic status, and public health programs and policies are also critical factors for sarcoidosis-related mortality [22,23,24,25,26,27,28,29]. Previous research showed sarcoidosis imposes a significant economic burden with considerable variations in healthcare spending on sarcoidosis patients in the US [7,30,31]. Baughman et al. indicated the median health care cost for patients with sarcoidosis was $18,663 per year but the yearly cost for the top 5% was $93,201 [7]. Similarly, Rice et al., in their study, found the average annual healthcare costs of high-cost sarcoidosis patients were 10 times that of the low-cost patients ($73,345 vs. $7,073) [31]. The average annual healthcare costs were even higher for patients in the top 2–5% and top 1%, at $119,878 and $375,436, respectively. Their study also identified several high-cost indicators, including complicated patients with higher rates of comorbidities, increased health care resource use (hospitalizations and specialty visits), and advanced treatments with biologic therapies (adalimumab and/or infliximab) [31]. Interestingly, Harper et al. recently also suggested low income was a leading predictor of poor outcome for sarcoidosis [32]. All these factors can be the reason that sarcoidosis-related AAMR in all populations was significantly associated with Overall and All Determinants health rankings.

There is a paradoxical association of state health rankings with sarcoidosis-related AAMR in European Americans. In our study, the Northeastern states have better health rankings, yet paradoxically higher mortality rates. One possible explanation is the heterogeneity in disease presentation and severity among different ethnic-racial backgrounds and the inconsistency of health care ability to identify the disease in different states. Establishing a definitive diagnosis for sarcoidosis can be challenging. Diagnosis of sarcoidosis is based on specific radiographic and pathologic findings in the setting of a compatible clinical presentation and exclusion of alternative diagnoses. A number of studies indicated European Americans tend to present with asymptomatic and chronic disease compared to African Americans [33,34,35]. This suggests states with better health rankings may have greater medical resources and abilities to identify chronic or asymptomatic patients, contributing to higher incidence, prevalence, and mortality rates for sarcoidosis. Diagnostic capacities and reporting sarcoidosis as primary or secondary causes of mortality is another major factor that might play a role.

FSR-WASOG Sarcoidosis Clinics are certified institutions that provide multidisciplinary care to patients with all forms of sarcoidosis. Among 50 states and 9 geographic divisions, New York State and South Atlantic have the largest numbers of FSR-WASOG Sarcoidosis Clinics (6 and 13, respectively). In the US, most Sarcoidosis Clinics are affiliated with university hospitals and built based on patients’ needs. Therefore, the law of supply and demand is a key point that could explain the relationship between Sarcoidosis Clinics numbers and sarcoidosis-related AAMR, suggesting states with Sarcoidosis Clinics have higher AAMR in all population compared to those without Sarcoidosis Clinics and divisions with higher AAMR tend to have larger numbers of Sarcoidosis Clinics for all population and African Americans. Sarcoidosis Clinics visit advanced and complicated patients that are at higher risk of mortality.

There are some strengths and limitations to our study. The standardized methods and measures using national population-based surveillance and relatively easy website-based analysis tool to obtain the underlying cause of death data and state health rankings in the US are our strengths [5,6]. However, the mortality data are based on death certificates from the National Center for Health Statistics and there are intrinsic concerns and limitations related to the inconsistent report for the cause of death by clinicians, misclassification, and incorrect coding of diagnoses in the dataset [3,4,5,6]. Another limitation is the use of state health rankings and numbers of Sarcoidosis Clinics to represent the health care disparities in the US may not be specific. Although America’s Health Rankings composes of five core measures (Behaviors, Community & Environment, Policy, Clinical Care, and Outcomes) to evaluate the health care status of each state, there are still some crucial parameters not included. We did not have information for sarcoidosis incidence, prevalence, diagnostic capacity, disease severity, therapeutic resources, and comorbidities by state, which may play crucial roles in explaining the geographic differences of mortality rates. Therefore, further research and investigations are warranted to support our findings.

Overall, our study showed clear racial and geographic differences in sarcoidosis-related AAMR in the US and its marked association with state health rankings and the numbers of Sarcoidosis Clinics. The current study suggests sarcoidosis-related AAMR could be an indicator to reflect the health care disparities between states and the improvement of overall health status in the US may also improve the mortality rates of sarcoidosis.

## Figures and Tables

**Figure 1 jcm-10-02366-f001:**
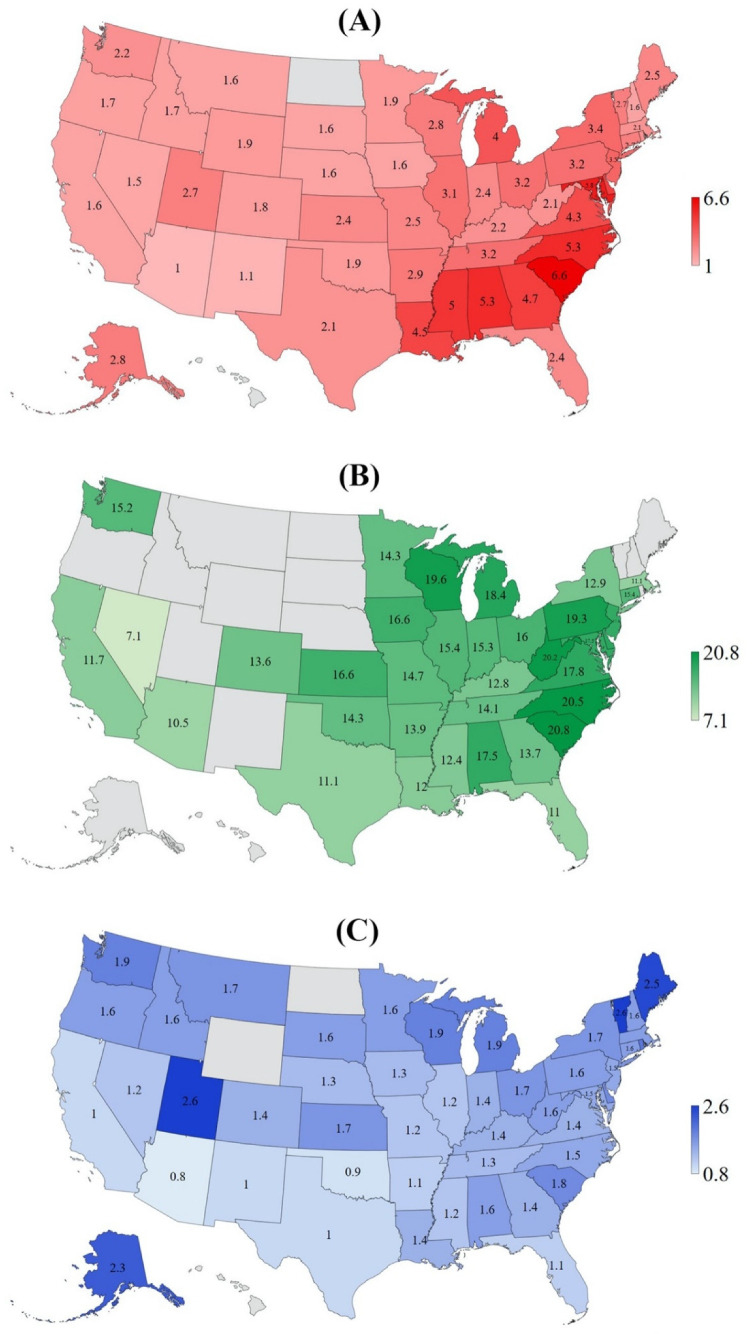
Map of Sarcoidosis-Related AAMR (age-adjusted mortality rate) per 1,000,000 for (**A**) All Population, (**B**) African Americans, and (**C**) European Americans for 50 States of the US, 1999 to 2018.

**Figure 2 jcm-10-02366-f002:**
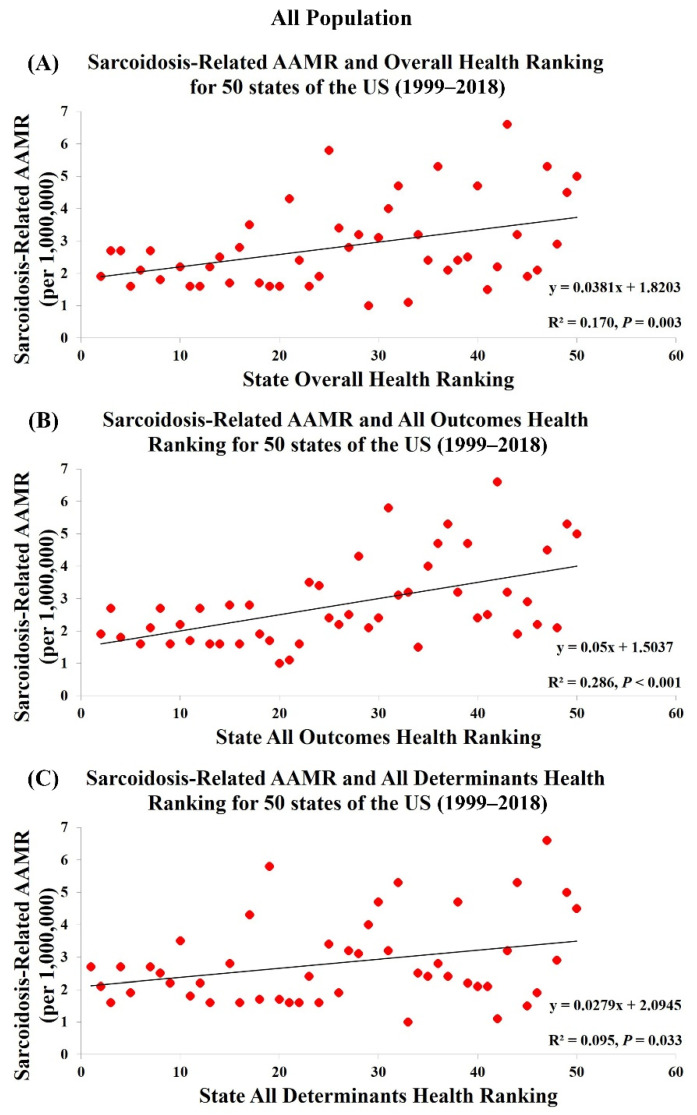
Sarcoidosis-Related AAMR in All Population are significantly associated with (**A**) Overall Health Rankings, (**B**) All Outcomes Health Rankings, and (**C**) All Determinants Health Rankings.

**Figure 3 jcm-10-02366-f003:**
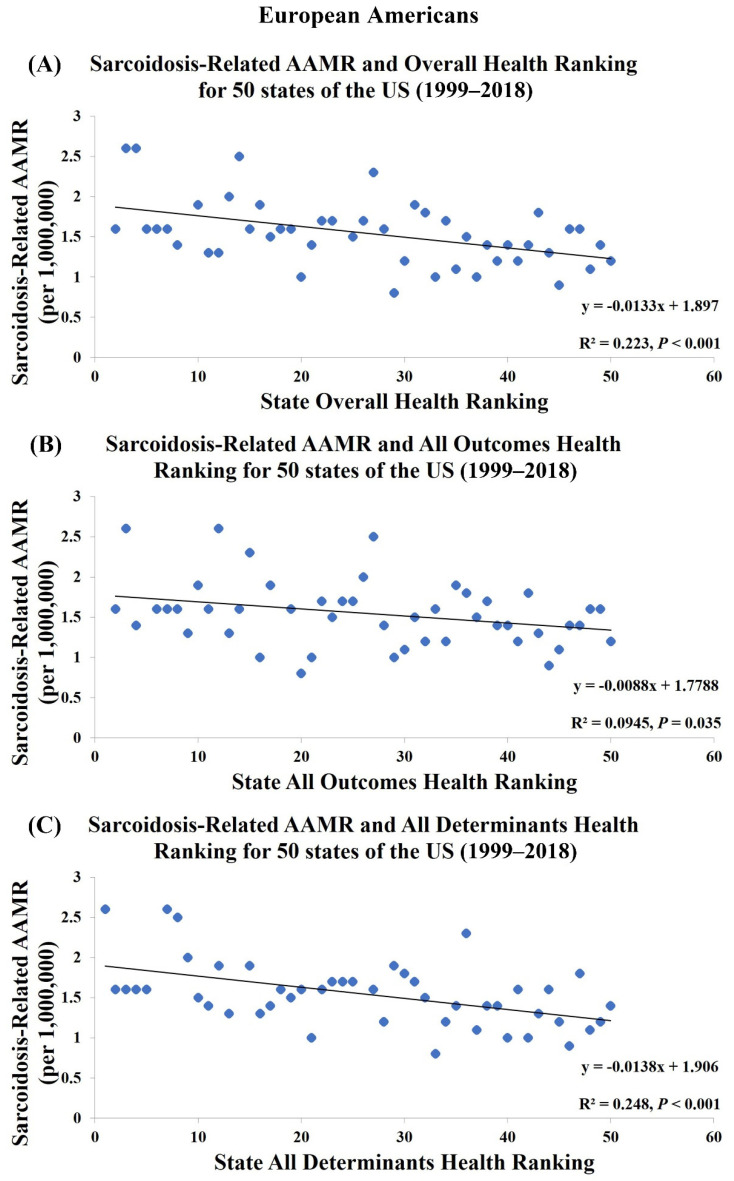
Sarcoidosis-Related AAMR in European Americans is significantly associated with (**A**) Overall Health Rankings, (**B**) All Outcomes Health Rankings, and (**C**) All Determinants Health Rankings.

**Figure 4 jcm-10-02366-f004:**
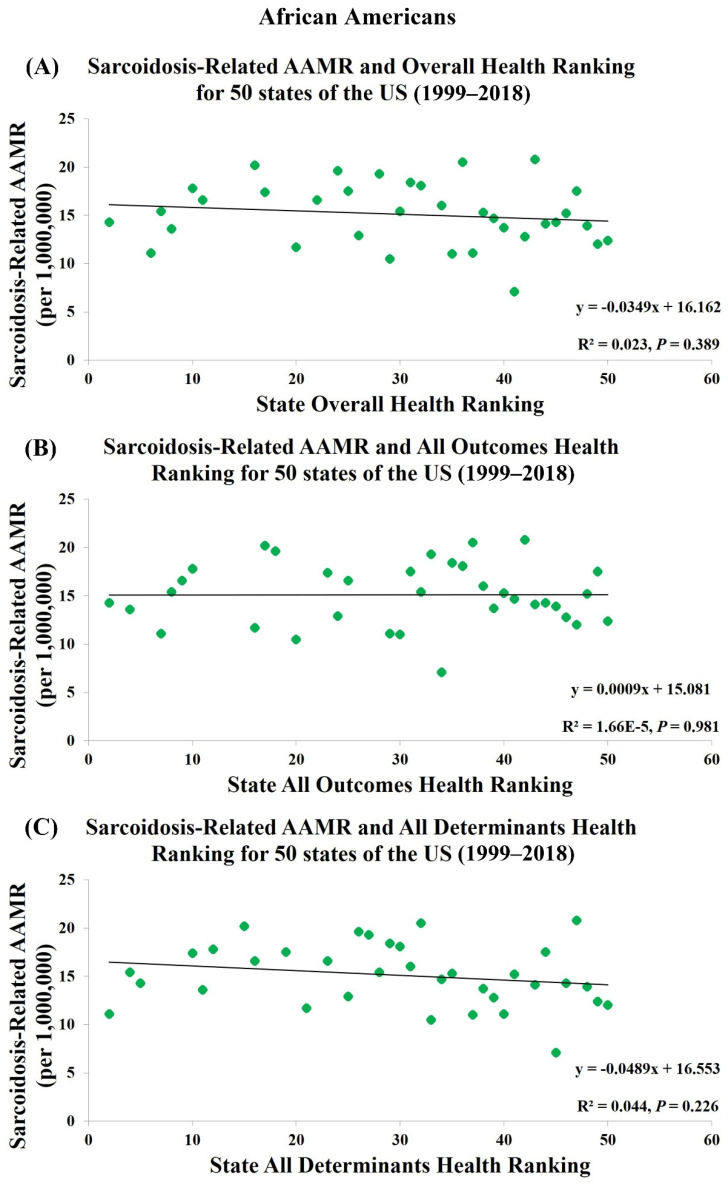
Sarcoidosis-Related AAMR in African Americans is not associated with (**A**) Overall Health Rankings, (**B**) All Outcomes Health Rankings, and (**C**) All Determinants Health Rankings.

**Table 1 jcm-10-02366-t001:** Summary of State Health Rankings, Sarcoidosis-Related Age-Adjusted Mortality Rate, and FSR-WASOG (Foundation for Sarcoidosis Research-World Association for Sarcoidosis and Other Granulomatous Disorders) Sarcoidosis Clinics of 50 States.

State	1999–2018 America’s Health Rankings (AHR)			1999–2018 Age-Adjusted Mortality Rate (AAMR) Per 1,000,000			FSR-WASOG Sarcoidosis Clinics
	Overall	All Outcomes	All Determinants	All Population	African Americans	European Americans	
Hawaii	1	1	6	No Data	No Data	No Data	0
Minnesota	2	2	5	1.9	14.3	1.6	2
Vermont	3	12	1	2.7	No Data	2.6	0
Utah	4	3	7	2.7	No Data	2.6	1
New Hampshire	5	6	3	1.6	No Data	1.6	0
Massachusetts	6	7	2	2.1	11.1	1.6	2
Connecticut	7	8	4	2.7	15.4	1.6	1
Colorado	8	4	11	1.8	13.6	1.4	1
North Dakota	9	5	14	Unreliable	No Data	Unreliable	0
Washington	10	10	12	2.2	15.2	1.9	0
Iowa	11	9	16	1.6	16.6	1.3	1
Nebraska	12	13	13	1.6	No Data	1.3	0
Rhode Island	13	26	9	2.2	No Data	2.0	0
Maine	14	27	8	2.5	No Data	2.5	0
Idaho	15	11	20	1.7	No Data	1.6	0
Wisconsin	16	17	15	2.8	19.6	1.9	0
New Jersey	17	23	10	3.5	17.4	1.5	2
Oregon	18	19	18	1.7	No Data	1.6	0
South Dakota	19	14	22	1.6	No Data	1.6	0
California	20	16	21	1.6	11.7	1.0	3
Virginia	21	28	17	4.3	17.8	1.4	3
Kansas	22	25	23	2.4	16.6	1.7	1
Montana	23	22	24	1.6	No Data	1.7	0
Wyoming	24	18	26	1.9	No Data	Unreliable	0
Maryland	25	31	19	5.8	17.5	1.5	1
New York	26	24	25	3.4	12.9	1.7	6
Alaska	27	15	36	2.8	No Data	2.3	0
Pennsylvania	28	33	27	3.2	19.3	1.6	3
Arizona	29	20	33	1.0	10.5	0.8	2
Illinois	30	32	28	3.1	15.4	1.2	4
Michigan	31	35	29	4.0	18.4	1.9	3
Delaware	32	36	30	4.7	18.1	1.8	0
New Mexico	33	21	42	1.1	No Data	1.0	0
Ohio	34	38	31	3.2	16.0	1.7	3
Florida	35	30	37	2.4	11.0	1.1	3
North Carolina	36	37	32	5.3	20.5	1.5	3
Texas	37	29	40	2.1	11.1	1.0	4
Indiana	38	40	35	2.4	15.3	1.4	0
Missouri	39	41	34	2.5	14.7	1.2	1
Georgia	40	39	38	4.7	13.7	1.4	2
Nevada	41	34	45	1.5	7.1	1.2	0
Kentucky	42	46	39	2.2	12.8	1.4	0
South Carolina	43	42	47	6.6	20.8	1.8	1
Tennessee	44	43	43	3.2	14.1	1.3	2
Oklahoma	45	44	46	1.9	14.3	0.9	2
West Virginia	46	48	41	2.1	20.2	1.6	0
Alabama	47	49	44	5.3	17.5	1.6	1
Arkansas	48	45	48	2.9	13.9	1.1	0
Louisiana	49	47	50	4.5	12.0	1.4	2
Mississippi	50	50	49	5.0	12.4	1.2	0

**Table 2 jcm-10-02366-t002:** Summary of Sarcoidosis-Related Age-Adjusted Mortality Rate and FSR-WASOG Sarcoidosis Clinics of 9 Geographic Divisions.

Geographic Division	1999–2018 Age-Adjusted Mortality Rate (AAMR) per 1,000,000			FSR-WASOG Sarcoidosis Clinics
	All Population	African Americans	European Americans	
New England	2.3	12.4	1.8	3
Middle Atlantic	3.4	15.2	1.6	11
East North Central	3.2	16.5	1.6	10
West North Central	2.1	14.3	1.4	5
South Atlantic	4.2	16.3	1.3	13
East South Central	3.7	14.6	1.4	3
West South Central	2.5	11.7	1.1	8
Mountain	1.5	10.6	1.3	4
Pacific	1.6	12.1	1.2	3

## Data Availability

All relevant data are within the manuscript and its Appendix A.

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
