# Peer review of "State-Level Health Disparity Is Associated with Sarcoidosis Mortality"

_jcm, 2021, doi:10.3390/jcm10112366_

Round 1

Reviewer 1 Report

The manuscript is well written.    The findings are well discussed. The massage is very important.

   Minor edits are required

Author Response

Thank you for your time and comment.

Reviewer 2 Report

Lee et al. present in their manuscript a detailed analysis of sarcoidosis mortality in US and link it to geographic and health-care-associated surrogates. The manuscript is well written and is interesting for physicians with focus on sarcoidosis and / or health economics. I would like to mention a few points that could be addressed to improve the manuscript: 

  1. I‘m not familiar with the AAMR, but what I understood from the method section, MR of a certain cohort is adjusted for an estimated MR for 2000 „normal“ persons. Is this „control cohort“ the same for the analysis of AAMR of Americans, African Americans and European Americans? How the authors deal with potentially different MR for persons of different ancestry? Could this (partially) explain the higher mortality rate in AA sarcoidosis patients?
  2. Is the control cohort used for estimation of AAMR specific for each state or is it general cohort for the whole US?
  3. Figure 2 and 3 at a first glance seem to be similar. Maybe the authors could indicate more clearly, that Figure 2 is for the whole cohort and Figure 3 for the European Americans. 
  4. The findings with sarcoidosis-associated higher mortality in regions with higher numbers of WASOG-certified sarcoidosis clinics is interesting, however I have some concerns
    1. Are these certifications based on official certification processes reflecting details standard operation procedures, diagnostic-related and treatment-related pathways that guarantee a high treatment quality?
    2. Is sarcoidosis prevalence comparable between regions with higher numbers of WASOG-certified clinics reflecting better diagnosis?
    3. Is the disease severity comparable between regions and with higher and lower numbers of WASOG clinics? 
    4. In my opinion, the correlation with sarcoidosis clinics in this manuscript is not urgently needed. It does not contribute to the finding of heterogenous mortality and does not prove or exclude that certified clinics improve management of sarcoidosis patients.  Rather it could be the scope of another research project to dissect underlying reasons for these observations. However, in the manuscript these observations are somewhat out of focus. 

Author Response

I‘m not familiar with the AAMR, but what I understood from the method section, MR of a certain cohort is adjusted for an estimated MR for 2000 „normal“ persons. Is this „control cohort“ the same for the analysis of AAMR of Americans, African Americans and European Americans?

Response: Yes, the control cohort is the same for the analysis of AAMR of Americans, African Americans and European Americans.

 How the authors deal with potentially different MR for persons of different ancestry?

Response: We have only access to race. We were unable to perform further analysis on different ancestry on each race.

Is the control cohort used for estimation of AAMR specific for each state or is it general cohort for the whole US?

Response: Thanks for question, It’s the general cohort for the whole US.

Figure 2 and 3 at a first glance seem to be similar. Maybe the authors could indicate more clearly, that Figure 2 is for the whole cohort and Figure 3 for the European Americans.

Response: Thanks for important comment. We modified Figure 2, 3 and added Figure 4 for clarification.

The findings with sarcoidosis-associated higher mortality in regions with higher numbers of WASOG-certified sarcoidosis clinics is interesting, however I have some concerns

    1. Are these certifications based on official certification processes reflecting details standard operation procedures, diagnostic-related and treatment-related pathways that guarantee a high treatment quality?

Response:  In conjunction with WASOG, World Association for Sarcoidosis and Other Granualomas, FSR evaluated each program with serval criteria including quantity of patients, quality of sarcoidosis trained physicians and being known as “sarcoidologist” , having multidisciplinary team including pulmonary, cardiology, nephology, hepatology, and other related specialties (for more information please review www.UMsarcoidosis.com),  being active in research (clinical and translational studies) and being active in patients and physicians educations.

    1. Is sarcoidosis prevalence comparable between regions with higher numbers of WASOG-certified clinics reflecting better diagnosis?

Response:  Unfortunately, we don’t have prevalence data by region for sarcoidosis. We added the new content to the manuscript to address the limitation “We did not have information for sarcoidosis incidence, prevalence, diagnostic capacity, disease severity, therapeutic resources, and comorbidities by state, which may play crucial roles in explaining the geographic differences of mortality rates.”(Page 14, line 239-241)

    1. Is the disease severity comparable between regions and with higher and lower numbers of WASOG clinics? 

Response:  Unfortunately, we don’t have disease severity data by region for sarcoidosis. We added the new content to the manuscript to address the limitation “We did not have information for sarcoidosis incidence, prevalence, diagnostic capacity, disease severity, therapeutic resources, and comorbidities by state, which may play crucial roles in explaining the geographic differences of mortality rates.” (Page 14, line 239-241)

    1. In my opinion, the correlation with sarcoidosis clinics in this manuscript is not urgently needed. It does not contribute to the finding of heterogenous mortality and does not prove or exclude that certified clinics improve management of sarcoidosis patients.  Rather it could be the scope of another research project to dissect underlying reasons for these observations. However, in the manuscript these observations are somewhat out of focus. 

Response:  Thanks for the reviewer’s comment.  We think it is important to bring the point that WASOG clinics probably visiting complicated patients that have higher risk of mortality. We adjusted that discussion to bring this point.

Reviewer 3 Report

The manuscript “State-level health disparity is associated with sarcoidosis mortality” is an examination of the CDC sarcoidosis mortality data from 1999-2018.  The manuscript can be improved by addressing the following points.

Major

  • If the goal was to generate state level disparities data, it seems that the WASOG Sarcoidosis clinics should not be grouped in to 9 geographic divisions. In fact, I would foster the analysis to compare states with and without sarcoidosis clinics and you will find higher mortality where there are sarcoidosis clinics. This would bolster your points made on page 10 to suggest that the clinics are demand driven.
  • The lack of association of sarcoidosis clinic number and African American mortality remains unexplained.
  • It seems that the association between poverty and its associated impact on sarcoidosis incidence is not even mentioned. In that the environmental associations with sarcoidosis are likely multiple, and patients are diagnosed late in the course of disease, might the true prevalence be orders of magnitude higher for states with high mortality?  I wonder if a figure to demonstrate proposed linkage between AAMR and incidence, diagnostic capacity, therapeutic resources, and co-morbidities would be a theoretic model that could help focus the discussion.
  • Might there be linkage between the number of patients followed in a WASOG sarcoidosis clinic indexed to the state or regional population with outcomes? This data is collected by the sarcoidosis clinic program inside WASOG.
  • It seems that the most interesting aspect of the paper is the lack of association between AAMR and overall health rankings in African Americans. My bias would be to bring figure e2 into the main manuscript and relegate the Caucasian data to the repository.  Might there be a smoking message hidden in this observation?

Minor

  • There is general need for editing for English. Just an example is found on page 10 line 195. “In our study, the Northeastern states have better health rankings but also with higher mortality rates.” should be “In our study, the Northeastern states have better health rankings, yet paradoxically higher mortality rates.” A general reading suggests many opportunities for an improved message.

Author Response

Major

  • If the goal was to generate state level disparities data, it seems that the WASOG Sarcoidosis clinics should not be grouped in to 9 geographic divisions. In fact, I would foster the analysis to compare states with and without sarcoidosis clinics and you will find higher mortality where there are sarcoidosis clinics. This would bolster your points made on page 10 to suggest that the clinics are demand driven.
  • Response:  Thanks for comment. We performed the analysis to compare states with and without sarcoidosis clinics and found “States with Sarcoidosis Clinics had 0.9 per 1,000,000 higher AAMR compared to those without Sarcoidosis Clinics in all population (P=0.018), but there was no statistical difference in European Americans (P=0.08) and African Americans (P=0.88).” We added the new content to the manuscript. (Page 2, line 86, 95,96; Page 12, line 147-151; Page 13, line 224, 225)  
  • The lack of association of sarcoidosis clinic number and African American mortality remains unexplained.

Response:  Thanks for comment. Sarcoidosis Clinics numbers were associated with higher AAMR in all population and African Americans.

  • It seems that the association between poverty and its associated impact on sarcoidosis incidence is not even mentioned. In that the environmental associations with sarcoidosis are likely multiple, and patients are diagnosed late in the course of disease, might the true prevalence be orders of magnitude higher for states with high mortality?  I wonder if a figure to demonstrate proposed linkage between AAMR and incidence, diagnostic capacity, therapeutic resources, and co-morbidities would be a theoretic model that could help focus the discussion.

Response:  Unfortunately, we don’t have information for sarcoidosis incidence/prevalence, diagnostic capacity, therapeutic resources, and co-morbidities by state. We added the new content to the manuscript to address the limitation “We did not have information for sarcoidosis incidence, prevalence, diagnostic capacity, disease severity, therapeutic resources, and comorbidities by state, which may play crucial roles in explaining the geographic differences of mortality rates.” (Page 14, line 239-241) 

  • Might there be linkage between the number of patients followed in a WASOG sarcoidosis clinic indexed to the state or regional population with outcomes? This data is collected by the sarcoidosis clinic program inside WASOG.

Response:   Thanks for comment.  We checked WASOG website and were unable to find the information for the number of patients followed in a WASOG sarcoidosis clinic. We will contact WASOG or FSR to obtain the data for our follow up manuscript.

  • It seems that the most interesting aspect of the paper is the lack of association between AAMR and overall health rankings in African Americans. My bias would be to bring figure e2 into the main manuscript and relegate the Caucasian data to the repository.  Might there be a smoking message hidden in this observation?

Response:   Thanks for comment. We moved prior “figure e2” into the main manuscript and labelled as “Figure 4”.

Minor

  • There is general need for editing for English. Just an example is found on page 10 line 195. “In our study, the Northeastern states have better health rankings but also with higher mortality rates.” should be “In our study, the Northeastern states have better health rankings, yet paradoxically higher mortality rates.” A general reading suggests many opportunities for an improved message.

Response:  Thanks for comment. We carefully edited the manuscript.

Round 2

Reviewer 3 Report

All comments were addressed.